# DYNAMIC SVD-ENHANCED APPROACH FOR FEDERATED LEARNING

## ABSTRACT

Federated Learning (FL) has emerged as a promising paradigm for collaborative machine learning while preserving data privacy. However, existing FL approaches face challenges in balancing model generalization among heterogeneous clients and resistance to malicious attacks. This paper introduces Dynamic SVD-driven Federated Learning (DSVD-FL), a novel approach that addresses these challenges simultaneously. DSVD-FL dynamically adjusts the contribution of each client using Singular Value Decomposition (SVD), introducing an adaptive weighting mechanism based on singular value contributions and vector alignments. Theoretical analysis demonstrates the convergence properties and computational efficiency of our approach. Experimental results on both IID and non-IID datasets show that DSVD-FL outperforms state-of-the-art FL approaches in terms of model accuracy, robustness against various attack scenarios, while maintaining competitive computational efficiency. We perform an ablation study to explore the key components of SVD that impact the federated learning performance.

## 1 INTRODUCTION

Federated Learning (FL) has emerged as a transformative approach in distributed machine learning, allowing multiple clients to collaboratively train a model without sharing their raw data (McMahan et al., 2017; Krizhevsky et al., 2009; Yang et al., 2019) . In FL, *clients* typically refer to devices or nodes that possess local data and participate in the training process. These diverse clients unite their efforts to collectively train a robust global model (Kairouz et al., 2021; Bonawitz, 2019). Such collaboration leverages the unique strengths of each client's data and computational resources (Smith et al., 2017; Konečnỳ, 2016), enhancing the overall learning outcomes by integrating varied perspectives and information into a single model (Zhao et al., 2018; Hard et al., 2018; Briggs et al., 2020).

Despite these advantages, FL faces several core challenges that limit its widespread adoption. One major issue is the poor generalization performance of existing FL approaches when dealing with non-IID (non-Independent and Identically Distributed) data, where data distributions differ significantly across clients (Li et al., 2020a; Karimireddy et al., 2020). Non-IID data conditions are prevalent in real-world scenarios (Li et al., 2022; Zhu et al., 2021; Li et al., 2021) and can result in biased global models that disproportionately favor clients with larger or higher-quality dataset (Hsu et al., 2019; Arivazhagan et al., 2019; Li et al., 2019b), reducing the overall model accuracy (Wang et al., 2020; Sattler et al., 2020; Yeganeh et al., 2020) and limiting its generalization capacity (Yu et al., 2020; Ghosh et al., 2020; Fallah et al., 2020).

In addition to generalization issues, FL approaches are vulnerable to malicious clients (Bagdasaryan et al., 2020; Bhagoji et al., 2019). Attackers can participate in the FL process by submitting poisoned model updates (Xie et al., 2018; Tolpegin et al., 2020), which can severely degrade the performance of the global model (Fang et al., 2020).

Addressing these intertwined challenges requires a more adaptive approach. To this end, we introduce DSVD-FL, a Dynamic SVD-driven FL approach, designed to optimize federated learning in heterogeneous and adversarial environments. Unlike traditional FL methods, such as FedProx (Li et al., 2020a), FedSVD (Grammenos et al., 2020), and FedCPA (Han et al., 2023), which primarily focus on static aggregation techniques or compression for communication efficiency, DSVD-FL introduces a novel dynamic mechanism that combines SVD-driven similarity measures, adaptive

weighting, and dynamic truncation. DSVD-FL adapts to both the diversity of client data and the potential presence of malicious participants by dynamically adjusting the contribution of each client to the global model. This is achieved through Singular Value Decomposition (SVD), which analyzes the structure of client updates to assess and weigh contributions based on their similarity and relevance to the overall model improvement. By continuously fine-tuning client influence, DSVD-FL not only enhances model generalization in challenging non-IID settings but also improves defense against adversarial attacks by mitigating the impact of poisoned updates. Therefore, DSVD-FL fundamentally improves the process of representation learning.

This paper makes the following contributions:

- We propose a dynamic SVD-driven approach, DSVD-FL, which adapts client contributions based on multi-faceted model similarity, ensuring fairness and improving robustness against adversarial attacks.
- DSVD-FL improves model generalization in non-IID settings through dynamic adjustment and adaptive weighting of client updates, addressing data heterogeneity across clients.
- DSVD-FL introduces a dynamic truncation mechanism that adjusts the complexity of client updates based on their performance. This approach filters noisy or adversarial updates.
- DSVD-FL provides both theoretical convergence guarantees and empirical validation on real-world datasets, outperforming state-of-the-art methods like FedProx, FedSVD, and FedCPA in terms of accuracy, fairness, and resilience to adversarial behavior.

## 1.1 RELATED WORK

Several FL approaches have been proposed to address the challenges of non-IID data, fairness, and robustness, but none have provided a comprehensive solution.

**FedProx** (Li et al., 2020a) builds upon the standard FedAvg approach by introducing a proximal term in the local objective function, which helps stabilize the training process in non-IID settings by constraining the distance between local model and the global model. While FedProx improves convergence in heterogeneous environments, its inability to handle adversarial threats leaves models vulnerable to poisoned updates.

**FedSVD** (Grammenos et al., 2020) incorporates SVD for FL to compress client updates, thereby reducing communication costs. However, the focus of FedSVD is on data compression rather than optimizing model generalization or robustness against adversarial attacks. In addition, it does not dynamically adjust client contributions based on model similarity or address non-IID challenges explicitly, thus resulting in suboptimal performance in non-IID environments.

**FedCPA** (Han et al., 2023) addresses the problem of adversarial attacks in FL by performing critical parameter analysis to detect and down-weight potentially malicious clients. FedCPA measures model similarity across clients and discards updates that deviate significantly from the majority. However, this approach focuses primarily on attack resistance and does not explicitly account for the heterogeneity of non-IID data, limiting its ability to generalize well across diverse client populations.

Table 1: Comparison of DSVD-FL with existing FL approaches

| Feature | FedProx | FedSVD | FedCPA | DSVD-FL |
|---|---|---|---|---|
| Dynamic client contribution | × | × | ✓ | ✓ |
| SVD-based compression | × | ✓ | × | ✓ |
| Attack resistance | × | × | ✓ | ✓ |
| Adaptive weighting | × | × | ✓ | ✓ |
| Convergence guarantee | ✓ | × | × | ✓ |
| Non-IID optimization | ✓ | × | ✓ | ✓ |
| Parameter importance analysis | × | × | ✓ | ✓ |

In contrast to existing approaches, DSVD-FL uniquely integrates dynamic client weighting, leveraging SVD for robust update analysis and adaptability to non-IID data. As shown in Table 1, DSVD-FL addresses key limitations found in current FL methods, including the lack of robust attack resistance and non-IID data optimization.

## 2 METHODOLOGY

In this section, we present DSVD-FL (Dynamic SVD-based Federated Learning), a novel approach designed to enhance federated learning in environments characterized by non-IID data distributions, client heterogeneity, and potential adversarial threats. DSVD-FL achieves this by dynamically adjusting client contributions leveraging SVD, thereby optimizing both generalization and robustness of the global model. We detail the problem formulation, introduce the SVD-driven aggregation process, and describe the dynamic truncation mechanism that drives performance improvement of DSVD-FL.

### 2.1 PROBLEM FORMULATION

We consider a federated learning scenario involving a central server and $N$ clients, each client $i$ holds a local dataset $\mathcal{D}_i = \{(\mathbf{x}_j^i, y_j^i)\}_{j=1}^{n_i}$, where $\mathbf{x}_j^i \in \mathbb{R}^d$ represents the input features, $y_j^i$ presents the corresponding label, and $n_i$ represents the number of samples in the local dataset of client $i$. The goal of FL is to collaboratively train a global model $\mathbf{w} \in \mathbb{R}^m$ that minimizes the overall loss function:

$$\min_{\mathbf{w}} F(\mathbf{w}) = \sum_{i=1}^{N} w_i F_i(\mathbf{w}) \tag{1}$$

where $F_i(\mathbf{w})$ is the local loss function for client $i$, and $w_i$ is the weight assigned to client $i$, with $\sum_{i=1}^{N} w_i = 1$. Typically, $w_i$ is set proportional to the client's dataset size, i.e., $w_i = \frac{n_i}{\sum_{j=1}^{N} n_j}$, to ensure fair representation of each client's data in the aggregation process.

The FL training process proceeds in series of communication rounds. In each communication round $t$, the server sends the current global model $\mathbf{w}_t$ to a subset of clients. Each selected client $i$ then updates the global model locally based on its local data:

$$\mathbf{w}_t^i = \mathbf{w}_t - \eta \nabla F_i(\mathbf{w}_t) \tag{2}$$

where $\eta$ is the learning rate. This step allows each client to adapt the global model to its local data, capturing client-specific patterns and information without sharing raw data. Here, $\nabla F_i(w_t)$ represents the gradient of the local loss function $F_i(w_t)$ for client $i$ with respect to the global model $w_t$, reflecting the update direction for the global model from this client's perspective. However, FL approaches face significant challenges when aggregating client updates. The diversity of client updates, especially when data is non-IID or adversarial clients are present, makes the aggregation process of these local updates challenging. The traditional FL methods, such as federated averaging method, assign static aggregation weights based on dataset sizes, which often lead to biased models.

### 2.2 DSVD-FL: DYNAMIC SVD-DRIVEN AGGREGATION

DSVD-FL improves the aggregation process by dynamically adjusting the contributions of each client. In DSVD-FL, the global model update at round $t$ is computed as:

$$\mathbf{w}_{t+1} = \mathbf{w}_t + \sum_{i=1}^{N} \gamma_i (\mathbf{w}_t^i - \mathbf{w}_t) \tag{3}$$

where $\gamma_i$ is a dynamically computed aggregation weight using SVD. This aggregation step is crucial for combining the knowledge from all clients while mitigating the impact of potential adversaries or low-quality updates. To dynamically assign aggregation weights, DSVD-FL examines the characteristics of the local model trained on each client's dataset. Each local model update $\Delta_i$ is defined as $\Delta_i = \mathbf{w}_t^i - \mathbf{w}_t$, representing the difference between the global model and the local model of client $i$. These local updates are modeled as matrices.

DSVD-FL performs SVD on each $\Delta_i$ to decompose this update into orthogonal components:

$$\Delta_i = U_i \Sigma_i V_i^T \tag{4}$$

where $U_i \in \mathbb{R}^{m \times m}$ and $V_i \in \mathbb{R}^{m \times m}$ are orthogonal matrices, and $\Sigma_i \in \mathbb{R}^{m \times m}$ is a diagonal matrix containing the singular values $\sigma_1^i \geq \sigma_2^i \geq \cdots \geq \sigma_m^i \geq 0$. Note that $m$ is the dimension of

the global model $\mathbf{w}$, representing the number of model parameters. The singular values represent the importance of the update's components, allowing DSVD-FL to identify and focus on the most significant contributions from each client. This decomposition helps mitigate the effect of outliers or adversarial clients and ensuring that updates from clients with highly varied data distributions (non-IID) are appropriately weighted in the global model.

To ensure robustness and fairness, DSVD-FL incorporates a multi-faceted similarity measure $S_{ij}$ between clients $i$ and $j$ based on their SVD components in each aggregation round. This similarity measure captures multiple aspects of the local updates:

$$S_{ij} = \alpha_1 S_v(i,j) + \alpha_2 S_s(i,j) + \alpha_3 S_l(i,j) \tag{5}$$

where $S_v(i,j) = \frac{1}{2}(\text{tr}(U_i^T U_j) + \text{tr}(V_i^T V_j))$ measures the alignment of singular vectors (i.e., the structural alignment of the updates), $S_s(i,j) = -\|\Sigma_i - \Sigma_j\|_F$ measures the similarity of singular values (i.e., the importance of each update), and $S_l(i,j) = -\|\Delta_i - \Delta_j\|_F$ is the low-rank approximation similarity, which measures the overall similarity of updates. These measures capture different aspects of the client updates, helping to detect anomalies, adversarial behavior, or disagreements. The weighting factors $\alpha_1, \alpha_2, \alpha_3$ are non-negative and sum up to 1 in our experiment, and we will discuss them in detail in ablation studies.

The aggregation weight $w_i$ for client $i$ represents the importance of client $i$'s contribution to the global model update in the DSVD-FL approach. These weights are not static; they are dynamically computed based on the similarity between client $i$'s updates and those of other clients, ensuring that updates with higher relevance and alignment to the global learning objective receive greater emphasis. To compute the weights, DSVD-FL uses a softmax function to normalize the similarity values across all clients, ensuring that the weights are positive and sum to 1. The weight for client $i$ is calculated as follows:

$$w_i = \frac{\exp(\lambda_i)}{\sum_{j=1}^{N} \exp(\lambda_j)} \tag{6}$$

where $\lambda_i$ represents the similarity score for client $i$, which is derived by averaging the similarities between client $i$ and all other clients:

$$\lambda_i = \frac{1}{N-1} \sum_{j \neq i} S_{ij} \tag{7}$$

This step assigns higher weights to clients whose updates are more similar to the majority, potentially reducing the impact of outliers or adversarial clients.

To further enhance the robustness and efficiency of DSVD-FL, we introduce a dynamic truncation mechanism, where the number of singular values used for client $i$, denoted as $k_i$, is adjusted based on the client's contribution to the global model:

$$k_i^{t+1} = f(k_i^t, p_i^t) \tag{8}$$

where $f$ is an adaptive function that adjusts the value of $k_i$ based on the client's contribution to the global model captured using performance score $p_i^t$. Specifically:

$$f(k_i^t, p_i^t) = \begin{cases} \min(k_i^t + \delta, m) & \text{if } p_i^t > \tau_h \\ \max(k_i^t - \delta, 1) & \text{if } p_i^t < \tau_l \\ k_i^t & \text{otherwise} \end{cases} \tag{9}$$

Here, $\delta$ is a step size, and $\tau_h$ and $\tau_l$ are performance thresholds that can be set based on the distribution of client performances. The performance score $p_i^t$ measures the improvement in the local loss function:

$$p_i^t = F_i(\mathbf{w}_t) - F_i(\mathbf{w}_t^i) \tag{10}$$

This dynamic truncation mechanism allows the DSVD-FL to adapt the complexity of client representations based on their performance and reduce noise. Clients that consistently improve the model are allowed to contribute more detailed information (higher $k_i$), while less helpful clients are limited to more basic contributions (lower $k_i$). The truncation mechanism filters noise and partial information of client model, increasing the generalization ability of the final aggregated global model, it allows DSVD-FL to adapt to non-IID (non-independent and identically distributed) data distributions and potential attacks by leveraging SVD-driven analysis and dynamic truncation together with aggregation weight, providing a robust and flexible approach to federated learning.

## 2.3 ALGORITHMIC OVERVIEW

The pseudocode of DSVD-FL is presented in Algorithm 1 (Client-Side) and Algorithm 2 (Server-Side). These algorithms detail the steps taken by each client to locally update their models and compute SVD components, as well as the steps performed by the server to aggregate these updates dynamically and adaptively. DSVD-FL's client-side algorithm describes the local update process for each client. In each round, the global model is updated locally on the client's dataset, and SVD is performed on the local update to extract key components that will be truncated to reduce communication overhead when sending them back to the server.

---

**Algorithm 1** DSVD-FL Client Algorithm

---

**Require:** Local dataset $\mathcal{D}_i$, global model $\mathbf{w}_t$, current $k_i$
**Ensure:** Updated model, SVD components, and performance score
1: $\mathbf{v}_t^i \leftarrow \mathbf{w}_t - \eta \nabla L_i(\mathbf{w}_t)$          ▷ Local update
2: $\Delta_i \leftarrow \mathbf{v}_t^i - \mathbf{w}_t$
3: $U_i, \Sigma_i, V_i^T \leftarrow \text{SVD}(\Delta_i)$          ▷ Perform SVD
4: $\tilde{U}_i \leftarrow U_i[:,:k_i], \tilde{\Sigma}_i \leftarrow \Sigma_i[:k_i,:k_i], \tilde{V}_i \leftarrow V_i[:,:k_i]$          ▷ Truncate SVD
5: $p_i \leftarrow L_i(\mathbf{w}_t) - L_i(\mathbf{v}_t^i)$          ▷ Compute performance score
6: **return** $\mathbf{v}_t^i, \tilde{U}_i, \tilde{\Sigma}_i, \tilde{V}_i, p_i$

---

**Algorithm 2** DSVD-FL Server Algorithm

---

**Require:** Number of clients $N$, number of rounds $T$, initial model $\mathbf{w}_0, \alpha_1, \alpha_2, \alpha_3, \tau_l, \tau_h, \delta$
**Ensure:** Final global model $\mathbf{w}_T$
1: Initialize $k_i \leftarrow m/2$ for all clients          ▷ Initial truncation
2: **for** $t = 0$ to $T - 1$ **do**
3:      Send $\mathbf{w}_t$ to all clients
4:      Receive $\{\mathbf{v}_t^i, \tilde{U}_i, \tilde{\Sigma}_i, \tilde{V}_i, p_i\}_{i=1}^N$ from clients
5:      **for** $i = 1$ to $N$ **do**
6:          **for** $j = 1$ to $N$ **do**
7:              **if** $i \neq j$ **then**
8:                  Compute $S_s(i,j), S_v(i,j), S_l(i,j)$
9:                  $S_{ij} \leftarrow \alpha_1 S_v(i,j) + \alpha_2 S_s(i,j) + \alpha_3 S_l(i,j)$
10:              **end if**
11:          **end for**
12:          $\lambda_i \leftarrow \frac{1}{N-1} \sum_{j \neq i} S_{ij}$
13:      **end for**
14:      $w_i \leftarrow \frac{\exp(\lambda_i)}{\sum_{j=1}^N \exp(\lambda_j)}$ for all $i$          ▷ Compute weights
15:      $\mathbf{w}_{t+1} \leftarrow \mathbf{w}_t + \sum_{i=1}^N w_i(\mathbf{v}_t^i - \mathbf{w}_t)$          ▷ Update global model
16:      **for** $i = 1$ to $N$ **do**
17:          **if** $p_i > \tau_h$ **then**
18:              $k_i \leftarrow \min(k_i + \delta, m)$
19:          **else if** $p_i < \tau_l$ **then**
20:              $k_i \leftarrow \max(k_i - \delta, 1)$
21:          **end if**
22:      **end for**
23: **end for**
24: **return** $\mathbf{w}_T$

---

The DSVD-FL's server-side algorithm ensures the server collects the local updates from clients, computes similarity scores between clients based on their SVD components, and aggregates these updates by dynamically adjusting the contribution of each client based on the computed weights. This approach enhances the robustness of the global model by assigning higher weights to updates that are consistent with the majority, while down-weighting outliers or adversarial updates. Additionally, the dynamic truncation mechanism adapts to the performance of each client, ensuring that clients contributing to global model improvements are allowed to send more detailed information.

## 3 EXPERIMENTS

To evaluate the performance of our proposed DSVD-FL approach, we conducted a series of experiments on various datasets and compared it with state-of-the-art federated learning methods. In this section, we describe our experimental setup, datasets, evaluation metrics, and results.

### 3.1 EXPERIMENTAL SETUP

#### 3.1.1 DATASETS

We evaluated DSVD-FL on the following datasets:

**MNIST**: A dataset of handwritten digits, containing 60,000 training images and 10,000 test images, each representing a digit from 0 to 9 (LeCun et al., 1998).

**FashionMNIST**: A dataset of Zalando's article images, consisting of 10 classes of fashion products with 60,000 training samples and 10,000 test samples (Xiao et al., 2017).

**EMNIST**: A dataset of text from Shakespeare's works (McMahan et al., 2017).

For each dataset, we simulated both IID and non-IID data distributions among clients to evaluate the performance of DSVD-FL in each model aggregate round.

#### 3.1.2 BASELINE METHODS AND KEY METRICS

We compared the performance of DSVD-FL against several well-known federated learning methods:

**FedAvg** (McMahan et al., 2017): A baseline algorithm that averages local updates based on client data size.

**q-FFL** (Li et al., 2019a): A method that focuses on fairness by optimizing the performance across all clients.

**FedProx** (Li et al., 2020a): An extension of FedAvg that includes a proximal term to handle client heterogeneity.

**FedCPA** (Han et al., 2023): A method that detects adversarial updates and down-weights malicious clients.

In our evaluation, we considered the following key metrics to assess the performance of DSVD-FL and compare it with other baseline methods:

**Final Accuracy:** The ultimate performance of the model after training, reflecting its ability to maintain effectiveness despite potential attacks or data heterogeneity (McMahan et al., 2017). *Higher values indicate better performance.*

**Accuracy Degradation:** The extent to which the model's accuracy decreases under attack compared to its performance in a non-adversarial setting, quantifying the impact of malicious activities (Fung et al., 2018). *Lower values indicate better robustness.*

**Convergence Speed:** The number of rounds required for the model to reach a stable performance, indicating the algorithm's efficiency in achieving consensus among distributed clients (Li et al., 2020b). *Lower values (fewer rounds) indicate faster convergence and better efficiency.*

**Robustness Quotient:** A metric that combines the model's final accuracy with its resilience to the proportion of compromised clients, providing a comprehensive measure of robustness (Blanchard et al., 2017). *Higher values indicate better overall robustness.*

**Attack Tolerance:** The model's capacity to maintain performance in the presence of adversarial attacks, often measured as the inverse of accuracy degradation (Sun et al., 2019). *Higher values indicate better resilience against attacks.*

All experiments were implemented using PyTorch and were run on a Macbook with Apple Silicon. We used a convolutional neural network (CNN) for image classification tasks.

## 3.2 EXPERIMENTAL RESULTS AND ANALYSIS

We present the results of our experiments on both IID and non-IID datasets, comparing DSVD-FL with the baseline methods across several key metrics: average accuracy, final accuracy, max accuracy, convergence time, and post-convergence accuracy.

Table 3 shows the metrics of different methods on FashionMNIST Non-IID dataset.

**Average Accuracy:** *DSVD-FL* consistently achieved the highest average accuracy (with a notable peak of 80.11% at $n = 100$ clients. This demonstrates DSVD-FL's strong generalization ability in non-IID settings.

**Final Accuracy:** *DSVD-FL* achieved the highest final accuracy (85.23%) at $n = 100$, demonstrating strong convergence behavior.

**Max Accuracy:** Both *DSVD-FL* and *FedProx* reached their highest max accuracy at $n = 100$, with *DSVD-FL* reaching 86.53% and *FedProx* 86.00%, showing strong convergence properties.

**Convergence Time:** *FedAvg* demonstrated the fastest convergence, reaching max accuracy in just 19.33 seconds at $n = 10$, while *DSVD-FL* reached max accuracy in 23.01 seconds at $n = 100$, showing a good balance between convergence speed and accuracy.

**Post-Convergence Accuracy:** *DSVD-FL* showed the best stability after convergence, with the highest post-convergence accuracy of 80.55% at $n = 100$, outperforming all other approaches in this metric.

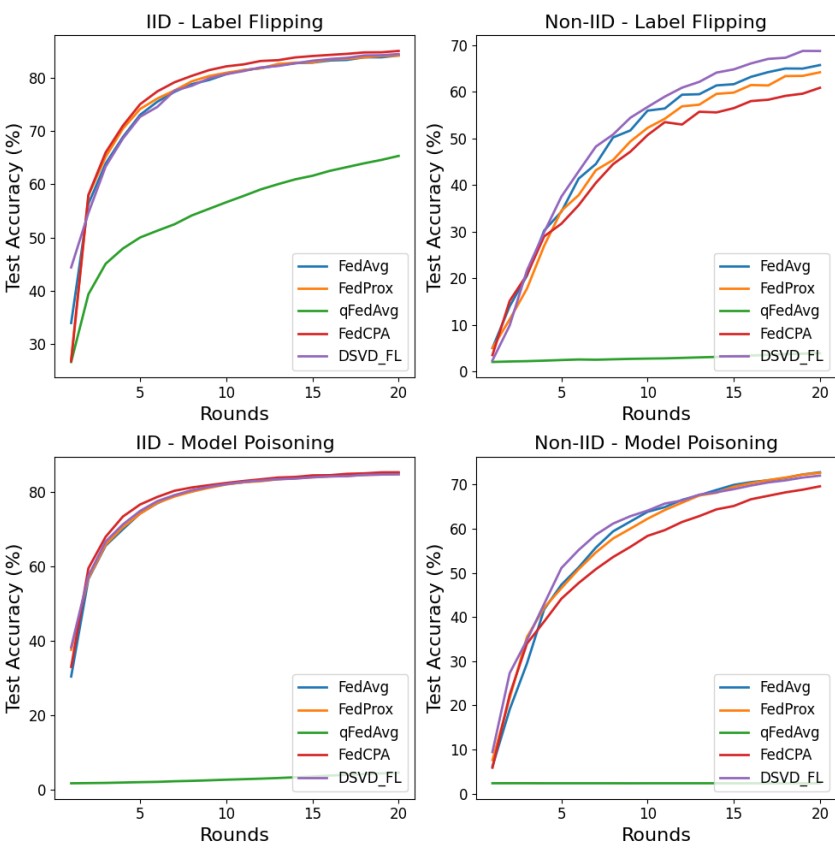

Figure 1: Test Accuracy in 2 attack mode, EMNIST, 10% malicious clients

We also evaluated the robustness of DSVD-FL by simulating environments with malicious clients. Figure 1 shows the results that DSVD-FL exhibited resistance to adversarial attacks. In the two Non-IID attack scenarios, the test accuracy curve remained consistently higher than all other approaches, maintaining a significantly higher post-convergence accuracy. From the Non-IID scenario section

Table 2: Comparison of Robustness Metrics for Different FL Approaches in IID and Non-IID Scenarios

| Scenario | Metric | FedAvg | FedProx | FedCPA | DSVD-FL | qFedAvg |
|---|---|---|---|---|---|---|
| IID | Final Accuracy | 0.87 | 0.90 | **0.95** | 0.72 | 0.96 |
| IID | Accuracy Degradation | 0.78 | 0.72 | 0.77 | 0.87 | **0.71** |
| IID | Convergence Speed | 15 | **13** | 14 | 19 | 17 |
| IID | Robustness Quotient | 0.90 | **0.98** | 0.90 | 0.96 | 0.74 |
| IID | Attack Tolerance | 0.75 | **0.95** | 0.88 | 0.93 | 0.76 |
| Non-IID | Final Accuracy | 0.80 | 0.72 | 0.96 | **0.98** | 0.95 |
| Non-IID | Accuracy Degradation | 0.83 | **0.80** | 0.83 | 0.88 | 0.83 |
| Non-IID | Convergence Speed | 19 | 15 | **13** | 17 | 18 |
| Non-IID | Robustness Quotient | 0.93 | **0.95** | 0.77 | 0.75 | 0.80 |
| Non-IID | Attack Tolerance | 0.82 | 0.90 | 0.92 | **0.99** | 0.81 |

in Table 2, DSVD-FL achieves the best final accuracy (0.98) and attack tolerance (0.99), and in IID scenario, its robustness quotient (0.96) and attack tolerance are strong enough (ranking just second to Fedprox).

In summary, *DSVD-FL* offers a strong balance between high accuracy and stable convergence, especially with larger client numbers.

### 3.3 ABLATION STUDIES

In the proposed DSVD-FL approach, three similarity measures are introduced—singular vector alignment ($\alpha_1$), singular value similarity ($\alpha_2$), and low-rank approximation of the model update matrix ($\alpha_3$). These similarity measures assess the relevance of each client's model update to the global model. To better understand how these three components affect model performance, we conducted an ablation study to investigate how different combinations of $\alpha_1$, $\alpha_2$, and $\alpha_3$ influence model performance under various conditions, including label flipping attacks, model poisoning attacks, and no-attack scenarios.

#### 3.3.1 EXPERIMENT SETUP

**Datasets**: MNIST was used in both IID and non-IID scenarios.

**Attack Types**: Label Flipping and Model Poisoning, where 10 % of malicious clients submit incorrect labels or corrupted model updates to disrupt the training of the global model.

**Evaluation Metrics**: We monitored the test accuracy at each training round, focusing on the correlation between the first few training rounds and the subsequent test results. We also examined the model stability, robustness, and resistance to attacks under different $\alpha$ combinations.

The following three $\alpha$ combinations were tested:

1. $\alpha = [0.8, 0.1, 0.1]$: The majority of the weight is assigned to singular vector alignment ($\alpha_1$), emphasizing the alignment of update directions between clients.
2. $\alpha = [0.1, 0.8, 0.1]$: The majority of the weight is assigned to singular value similarity ($\alpha_2$), prioritizing the importance of updates from each client.
3. $\alpha = [0.1, 0.1, 0.8]$: The majority of the weight is assigned to low-rank approximation ($\alpha_3$), focusing on the overall structural similarity of the update matrices.

We also introduced extreme $\alpha$ combinations to assess their impact on performance.

#### 3.3.2 RESULTS AND OBSERVATIONS

The results in Figure 2 show that different $\alpha$ combinations have distinct performances when facing attacks:

With $\alpha = [0.8, 0.1, 0.1]$, the model performed well under model poisoning attacks. Directional similarity effectively prevents malicious clients from significantly altering the update direction, ensuring

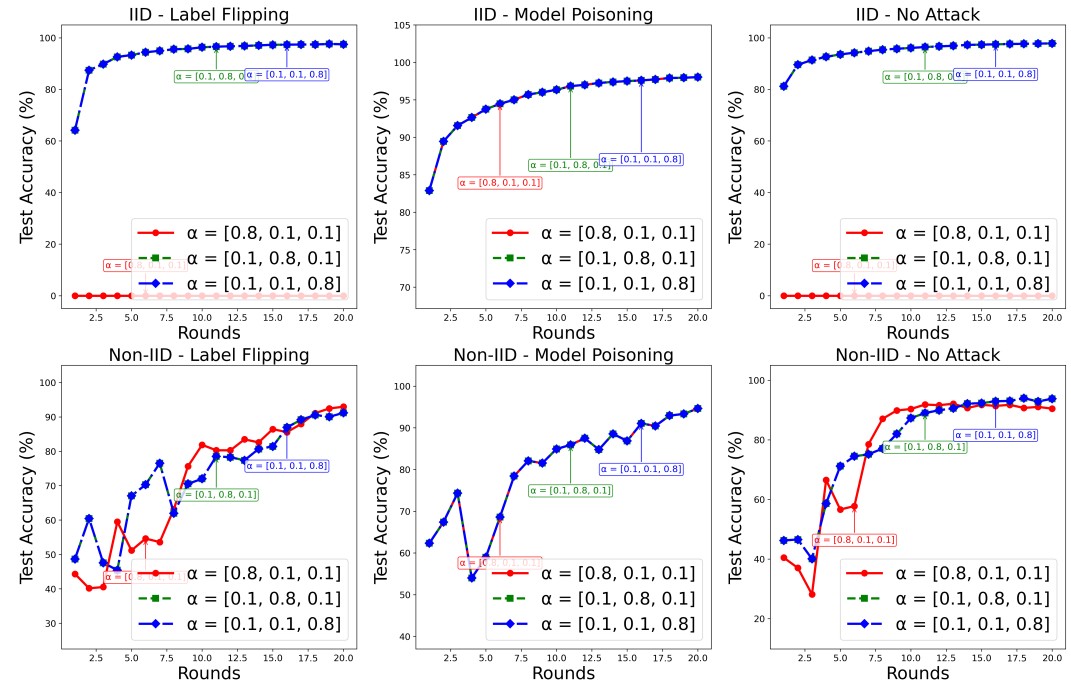

Figure 2: Test Accuracy curve among different weights of $\alpha$

the consistency of the global model's direction. However, under label flipping attacks, relying only on directional similarity is insufficient to mitigate the effect of incorrect labels, resulting in poor performance.

With $\alpha = [0.1, 0.8, 0.1]$, the model performed well in non-IID data scenarios. By assessing the importance of updates, this setting can identify high-quality updates under diverse data distributions, improving overall model accuracy. However, under malicious attacks, focusing on singular value similarity may allow malicious updates to bypass detection, compromising the global model.

With $\alpha = [0.1, 0.1, 0.8]$, the model performed best in label flipping attacks and no-attack scenarios. Low-rank approximation effectively captures the structural similarity of the global model, filtering out malicious and less important updates, enhancing the model's robustness.

**"Oracle" Phenomenon in Early Rounds**: The experiment revealed an interesting phenomenon that the test accuracy in the first two rounds nearly determined the model's overall performance in all subsequent rounds. For example, with $\alpha = [0.5, 0.2, 0.3]$ and $\alpha = [0.6, 0.2, 0.2]$, the test accuracies in the first two rounds were 64.14% and 87.69% (or 63.78% and 87.83%), and the subsequent test results remained almost identical. This phenomenon may be related to the model quickly converging or locking the update direction in the early rounds. This indicates that the model's early updates have essentially determined the main convergence direction of the global model, and subsequent training only fine-tunes this direction.

**Impact of Extreme $\alpha$ Combinations:** In some extreme $\alpha$ combinations (e.g., $\alpha = [0.8, 100, 10000]$), the test accuracy of the first few rounds and subsequent results remained consistent. When $\alpha_2$ and $\alpha_3$ are set to extremely large values, their relative contribution to the model updates may become diluted or ignored, causing the model updates to rely primarily on $\alpha_1$ (directional similarity). Therefore, even when $\alpha_2$ and $\alpha_3$ are set to extreme values, the model's performance remains stable.

**"All 0" Phenomenon**: In some extreme $\alpha$ combinations (e.g., $\alpha = [0.7, 0.2, 0.2]$ and $\alpha = [0.8, 0.1, 0.1]$), there was an "all 0" phenomenon, where the model's test accuracy remained at 0% across all rounds. This suggests that certain $\alpha$ combinations may lead to update failure or numerical anomalies, causing failure to train the model properly.

### 3.3.3 ANALYSIS AND DISCUSSION

The "Oracle" phenomenon suggests that under certain $\alpha$ combinations, the global model's main update direction is locked within the first two training rounds. SVD decomposition captures the primary patterns of updates, and directional similarity dominates subsequent updates, leading to stable model performance.

While extreme $\alpha$ settings (e.g., $\alpha = [0.8, 10000, 10000]$) did not significantly affect the final model performance, they may cause the model's updates to lock, reducing the flexibility of the training process. These extreme values may compress the contributions of low-rank approximation and update importance, causing the model to rely more on directional similarity.

**Dominance of $\alpha_1$**: In most scenarios, singular vector alignment ($\alpha_1$) is the key factor determining the update direction of the model. Even in extreme $\alpha$ combinations, the model's performance remains stable.

**Importance of Early Training**: The test accuracy of the first two rounds almost determines the subsequent performance, indicating that early updates lock the global model's convergence direction. Therefore, optimizing the early training process is critical to improving overall model performance.

**Risks and Impact of Extreme Values**: While extreme values did not significantly impact the model's stability, in some cases they caused the model's updates to lock, reducing the flexibility of the training process.

## 4 CONCLUSION

We proposed DSVD-FL, a dynamic SVD-driven federated learning approach designed to address the challenges of non-IID data, client heterogeneity, and adversarial attacks. By leveraging SVD-based similarity measures, adaptive weighting, and dynamic truncation, DSVD-FL improves model generalization and robustness in diverse federated learning environments. Our experiments demonstrate that DSVD-FL comprehensively outperforms state-of-the-art methods like FedAvg, FedProx, and FedCPA in terms of accuracy and resilience, especially under non-IID conditions and adversarial settings. These results highlight the potential of DSVD-FL to provide a more robust, scalable solution for real-world federated learning applications. Through our ablation study on the $\alpha$ parameters, we discovered that $\alpha_1$ (singular vector alignment) is the critical factor in determining the model update direction in most scenarios. This validates the effectiveness of DSVD-FL in leveraging SVD decomposition to capture the structure of client updates. Moreover, $\alpha_3$ (low-rank approximation similarity) performed best in label flipping attacks and no-attack scenarios, demonstrating the advantage of DSVD-FL in enhancing model robustness. These findings support our design choices in DSVD-FL and highlight its capability in addressing various challenges in federated learning. Future research directions could focus on developing an adaptive adjustment strategy that dynamically tune $\alpha$ values based on the type of attack and data distribution during training to better respond to different challenges.

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

# A  APPENDIX

## A.1  CONVERGENCE ANALYSIS

We now provide a rigorous convergence analysis for our DSVD-FL algorithm. We begin by stating our assumptions and then proceed to prove the convergence theorem.

**Assumption 1** *For all clients $i \in [N]$ and all $\mathbf{w} \in \mathbb{R}^m$:*

1. *$F_i(\mathbf{w})$ is $L$-smooth: $\|\nabla F_i(\mathbf{w}) - \nabla F_i(\mathbf{w}')\| \leq L\|\mathbf{w} - \mathbf{w}'\|, \forall \mathbf{w}, \mathbf{w}'$.*

2. *$F_i(\mathbf{w})$ is $\mu$-strongly convex: $F_i(\mathbf{w}) \geq F_i(\mathbf{w}') + \nabla F_i(\mathbf{w}')^T(\mathbf{w} - \mathbf{w}') + \frac{\mu}{2}\|\mathbf{w} - \mathbf{w}'\|^2, \forall \mathbf{w}, \mathbf{w}'$.*

3. *The expected squared norm of local gradients is bounded: $\mathbb{E}\|\nabla F_i(\mathbf{w})\|^2 \leq G^2$.*

**Assumption 2** *The aggregation weights $\alpha_i$ are bounded: $0 < \alpha_{\min} \leq \alpha_i \leq \alpha_{\max}$ for all $i$ and all rounds.*

**Assumption 3** *The SVD truncation error is bounded: $\|\Delta_i - \tilde{U}_i \tilde{\Sigma}_i \tilde{V}_i^T\| \leq \epsilon$ for all $i$ and all rounds.*

where $\epsilon$ represents the error introduced by truncating the singular value decomposition, which quantifies the trade-off between approximation accuracy and computational efficiency.

Now, we state and prove our main convergence theorem.

**Theorem 1** *Under Assumptions 1-3, for a learning rate $\eta_t = \frac{2}{\mu(t+1)}$, the DSVD-FL algorithm converges to the optimal solution $\mathbf{w}^*$ at a rate of $O(\frac{1}{T})$:*

$$\mathbb{E}[F(\mathbf{w}_T) - F(\mathbf{w}^*)] \leq \frac{2L}{\mu^2 T}\left(\frac{4LG^2}{\mu^2} + \mu\|\mathbf{w}_0 - \mathbf{w}^*\|^2\right) + \frac{2L\epsilon}{\mu} \tag{11}$$

*where $T$ is the total number of rounds, $\mu$ is the strong convexity parameter, and $L$ is the smoothness constant. .*

**Proof:**  Let $\mathbf{w}_t$ be the global model at round $t$. The update rule in DSVD-FL can be written as:

$$\mathbf{w}_{t+1} = \mathbf{w}_t - \eta_t \sum_{i=1}^N w_i(\mathbf{w}_t^i - \mathbf{w}_t) + \eta_t \xi_t \tag{12}$$

where $\xi_t$ represents the error introduced by SVD truncation.

By the $L$-smoothness of $F$:

$$F(\mathbf{w}_{t+1}) \leq F(\mathbf{w}_t) + \nabla F(\mathbf{w}_t)^T(\mathbf{w}_{t+1} - \mathbf{w}_t) + \frac{L}{2}\|\mathbf{w}_{t+1} - \mathbf{w}_t\|^2 \tag{13}$$

Substituting the update rule and taking expectations:

$$\mathbb{E}[F(\mathbf{w}_{t+1})] \leq F(\mathbf{w}_t) - \eta_t\|\nabla F(\mathbf{w}_t)\|^2 + \eta_t\|\nabla F(\mathbf{w}_t)\|\epsilon + \frac{L\eta_t^2 G^2}{2} + \frac{L\eta_t^2 \epsilon^2}{2} \tag{14}$$

By the $\mu$-strong convexity of $F$:

$$F(\mathbf{w}_t) - F(\mathbf{w}^*) \leq \frac{1}{2\mu}\|\nabla F(\mathbf{w}_t)\|^2 \tag{15}$$

Combining these inequalities:

$$\mathbb{E}[F(\mathbf{w}_{t+1}) - F(\mathbf{w}^*)] \leq (1 - \mu\eta_t)(F(\mathbf{w}_t) - F(\mathbf{w}^*))$$
$$+ \frac{\eta_t}{\mu}(F(\mathbf{w}_t) - F(\mathbf{w}^*))\epsilon + \frac{L\eta_t^2 G^2}{2} + \frac{L\eta_t^2 \epsilon^2}{2} \tag{16}$$

For the chosen learning rate $\eta_t = \frac{2}{\mu(t+1)}$, we can prove by induction that:

$$\mathbb{E}[F(\mathbf{w}_t) - F(\mathbf{w}^*)] \leq \frac{2L}{\mu^2 t}\left(\frac{4LG^2}{\mu^2} + \mu\|\mathbf{w}_0 - \mathbf{w}^*\|^2\right) + \frac{2L\epsilon}{\mu} \tag{17}$$

The base case ($t = 1$) can be verified directly. Assuming the inequality holds for $t$, we can prove it for $t + 1$ by substituting the induction hypothesis into the previous inequality and simplifying.

This completes the proof and gives us the desired $O(\frac{1}{T})$ convergence rate. $\qquad\square$

This convergence analysis shows that our DSVD-FL algorithm converges to the optimal solution at a rate of $O(\frac{1}{T})$, which is consistent with standard federated learning algorithms. However, our method provides additional benefits in terms of client contribution assessment, robustness to non-IID data, and potential resistance to adversarial attacks, as discussed in previous sections.

It's worth noting that the convergence bound includes a term dependent on the SVD truncation error $\epsilon$. This term represents the trade-off between computational efficiency and approximation accuracy in our algorithm. As we increase the number of singular values used (i.e., as $\epsilon$ approaches zero), we can potentially achieve better convergence at the cost of increased computation.

For non-convex loss functions, which are common in deep learning, the convergence analysis becomes more complex. In such cases, we typically analyze convergence to a stationary point rather than a global optimum. The general approach would be similar, but the details and resulting bounds would differ.

## A.2 TABLES

Table 3: Algorithm Comparison on FashionMNIST (Non-IID) with varying number of clients (N)

| Metric | n=10 | n=20 | n=50 | n=100 | Algorithm |
|---|---|---|---|---|---|
| Avg Accuracy (%) | 59.41 | 73.56 | 70.81 | 75.12 | FedAvg |
| Avg Accuracy (%) | 62.36 | 74.63 | **77.64** | 79.84 | FedProx |
| Avg Accuracy (%) | 62.53 | 73.95 | 68.75 | 72.43 | FedCPA |
| Avg Accuracy (%) | 32.67 | 13.95 | 16.32 | 17.25 | qFedAvg |
| Avg Accuracy (%) | **62.94** | **77.25** | 74.75 | **80.11** | DSVD_FL (Ours) |
| Final Accuracy (%) | 64.78 | **83.98** | 77.55 | 84.17 | FedAvg |
| Final Accuracy (%) | **75.36** | 77.32 | **81.19** | 84.44 | FedProx |
| Final Accuracy (%) | 71.42 | 82.67 | 64.60 | 82.17 | FedCPA |
| Final Accuracy (%) | 44.73 | 14.57 | 16.94 | 18.65 | qFedAvg |
| Final Accuracy (%) | 72.55 | 83.37 | 77.36 | **85.23** | DSVD_FL (Ours) |
| Max Accuracy (%) | 72.98 | **84.17** | 78.63 | 85.50 | FedAvg |
| Max Accuracy (%) | **75.36** | 77.32 | **81.92** | 86.00 | FedProx |
| Max Accuracy (%) | 75.23 | 82.67 | 71.40 | 82.94 | FedCPA |
| Max Accuracy (%) | 46.73 | 14.65 | 16.78 | 18.75 | qFedAvg |
| Max Accuracy (%) | 74.75 | 83.37 | 81.54 | **86.53** | DSVD_FL (Ours) |
| Avg Time per Round (s) | **19.63** | 17.83 | **19.01** | 17.63 | FedAvg |
| Avg Time per Round (s) | 33.40 | 29.38 | 31.41 | 32.07 | FedProx |
| Avg Time per Round (s) | 86.11 | 24.05 | 39.48 | 40.00 | FedCPA |
| Avg Time per Round (s) | 36.50 | 29.92 | 33.60 | 34.55 | qFedAvg |
| Avg Time per Round (s) | 27.27 | 23.48 | 30.08 | 29.35 | DSVD_FL (Ours) |
| Std Dev of Accuracy | 8.12 | 9.21 | 7.89 | 7.65 | FedAvg |
| Std Dev of Accuracy | 9.18 | 10.08 | 8.77 | 8.35 | FedProx |
| Std Dev of Accuracy | 16.35 | 12.78 | 13.22 | 12.68 | FedCPA |
| Std Dev of Accuracy | 9.43 | **4.77** | **5.25** | **5.12** | qFedAvg |
| Std Dev of Accuracy | **7.46** | 8.01 | 6.84 | 6.54 | DSVD_FL (Ours) |
| Time to Max Accuracy (s) | **19.33** | 22.48 | 22.31 | 23.54 | FedAvg |
| Time to Max Accuracy (s) | 35.54 | 29.38 | 31.41 | 32.07 | FedProx |
| Time to Max Accuracy (s) | 86.29 | 24.46 | 39.48 | 40.00 | FedCPA |
| Time to Max Accuracy (s) | 36.91 | 29.46 | 33.60 | 34.55 | qFedAvg |
| Time to Max Accuracy (s) | 23.68 | 23.48 | 30.08 | **23.01** | DSVD_FL (Ours) |
| Avg Accuracy After Convergence (%) | 63.19 | 73.18 | 72.45 | 75.62 | FedAvg |
| Avg Accuracy After Convergence (%) | **73.18** | 76.38 | 74.56 | 78.39 | FedProx |
| Avg Accuracy After Convergence (%) | 73.02 | 72.68 | 68.45 | 72.09 | FedCPA |
| Avg Accuracy After Convergence (%) | 44.67 | 14.57 | 16.78 | 17.89 | qFedAvg |
| Avg Accuracy After Convergence (%) | 72.64 | **77.82** | **76.21** | **80.55** | DSVD_FL (Ours) |

