# OpenReview forum: "Dynamic SVD-Enhanced Approach for Federated Learning"
_ICLR.cc/2025/Conference — ICLR 2025 Conference Withdrawn Submission_

### Official Review · Reviewer_ymBH · 2024-11-03

**Soundness:** 2
**Presentation:** 3
**Contribution:** 1
**Rating:** 3
**Confidence:** 5

**Summary:**

This paper focuses on the horizontal FL algorithm, specifically improving the robustness and byzantine resilience of the aggregation in FL. The paper decomposes the gradient with SVD and performs aggregation based on the similarities of the SVD between each pair of clients. Based on the similarity, the algorithm assigns different weights for model/update aggregation. Numerical results on three datasets (MNIST, Fashion MNIST, and Shakespeare) are used to evaluate the performance of the proposed aggregation approach on different evaluation matrices.

**Strengths:**

This paper proposes a novel aggregation approach for improving the robustness of FL in the non-IID data scenario.

The experiment setting is clearly described. The algorithm is also clearly described.

Extensive ablation studies are conducted to evaluate the performance of the proposed aggregation method.

**Weaknesses:**

1. Significance of the proposed method.
    1. From the numerical result 3.2, it is unclear whether the proposed algorithm outperforms the SOTA in terms of robustness and byzantine resilience. Figure 1 and table only show that the proposed method only outperforms other methods in two settings (non-IID label flipping accuracy and non-IID in Table 2), and in the IID case, it even has the worst accuracy. It is unclear why should we use the proposed method.
    2. On the robustness of the algorithm. The ablation study reports that the algorithm is sensitive to the choice of the $\alpha$'s, and in some cases, it even collapses. The author should provide a more detailed ablation study (grid search) on the combinations of these parameters since the current result does not provide any clear trend in the choice of parameters.

2. Lack of theoretical support.
    1. The authors fail to provide any theoretical analysis of the algorithm; either its stability or convergence analysis is missing, weakening its significance. The theoretical analysis in Appendix A.1 does not look correct to me. Specifically, where Assumption 2 is used is unclear; how eq(13) becomes eq(14), and becomes eq(16) are also unclear.
    2. More explanation of the intuition is required. For example, why $p_i^t, \delta$, and $f$ are used to adjust the rank of the SVD? Why is the softmax function used for weight normalization instead of other functions? Why specific $S_v, S_s, S_l$ are chosen while other distances are not used?
    3. Lacks of communication/computation complexity analysis. The author should discuss how much memory/communication is increased/reduced by using the SVD aggregation.

**Questions:**

Please address the weaknesses above.

---

### Official Review · Reviewer_xfHX · 2024-11-04

**Soundness:** 2
**Presentation:** 1
**Contribution:** 2
**Rating:** 3
**Confidence:** 4

**Summary:**

This paper proposes a federated learning method that dynamically adjusts client aggregation weights based on singular value decomposition (SVD). Before each round of aggregation, the model updates generated by individual clients undergo SVD, resulting in $M = U \Sigma V^*$. The method then computes the similarity between clients based on the singular value vector $\Sigma$ and the orthogonal matrices $U$ and $V$. Clients with higher similarity are assigned greater aggregation weights. The paper claims that this approach improves the generalization of federated learning under heterogeneous data, enhances fairness, and increases robustness against adversarial attacks.

**Strengths:**

The paper provides a convergence analysis.

**Weaknesses:**

- The paper claims that the proposed method improves fairness and model generalization. However, I did not find sufficient experimental results to support these claims. There is no report on the accuracy variance across clients, nor any experiments that reflect the generalization capability of the model.

- The number of baseline methods compared in the experiments is quite limited and somewhat outdated. The paper only compares against q-FFL (2019), FedProx (2020), and FedCPA (2023).

- DSVD-FL seems to incur higher communication overhead. A comparison of communication costs across methods should be provided.

- The paper does not specify the exact model architecture used. It merely states: "We used a convolutional neural network (CNN) for image classification tasks."

**Questions:**

How does determining aggregation weights based on client similarity improve fairness in federated learning? Would this not lead to a significant performance drop for clients with more unique characteristics?

---

### Official Review · Reviewer_PxwM · 2024-11-04

**Soundness:** 2
**Presentation:** 2
**Contribution:** 2
**Rating:** 5
**Confidence:** 4

**Summary:**

The paper studies the problem of federated model training in a heterogeneous setting where each client may have varying data distributions affecting their ability to converge to a single global model. By looking at similar client model updates based on an SVD-based similarity function, this article develops a method to leverage the most similar model updates in each round. Client updates that are the most similar overall tend to then have a higher averaging weight based on the proposed algorithm ensuring overall global convergence. Subsequently, the authors claim that this algorithm will ensure better resistance to attacks while maintaining a good convergence guarantee. The above claims are then verified by empirical results and a theoretical convergence guarantees.

**Strengths:**

- The paper proposes looking at pairwise client similarities (based on differences between the global model and local models and an SVD-based similarity function) to identify the client model updates that may be the most likely to be consistent with the expected global model update.
- The empirical results demonstrate the benefits of leveraging the proposed weighted scheme of preferring clients more similar to the majority of clients.
- Theoretical results provide the proposed method's convergence guarantees.

**Weaknesses:**

- The motivation behind leveraging SVD is unclear and requires more consideration in the write-up. For instance, if a cosine similarity-based function based on the gradients or the local model parameters of each client is used to gauge pair-wise client sameness then how does it affect the final result? Further, showing a result comparing such methods with the SVD approach would help cement the efficacy of the SVD approach.
- The presentation of algorithms 2 and 3 is inadequate and requires a better flow. It would be better to consider showcasing the step-by-step model training process while also motivating the need for the newer steps. For example, the roles of the performance score and thresholds are unclear.
- The presentation of the novelty of the method over FedProx is somewhat unclear besides the idea that outliers will not affect the model training as much based on the lower similarity score.
- Furthermore, consider that we have three groups of clients, with one majority class, one minority class, and one set of outlier clients. Further, suppose the outliers are more similar to the majority class than the minority class. Then, overall it seems the outliers will get a higher weight in the model updates than the minority class. In such a case, intuitively it seems the model training will be inferior to FedProx where all clients tend to grow close to the central model based on the proximal term. Can the authors discuss such a case in depth or provide relevant experimental results?
- Also, suppose a weighted FedProx is developed with client weights based on a similar pairwise client approach, then it seems that it would help avoid the issue above. Can the authors elaborate here?
- Finally, can the authors discuss the limitations of this method? It seems that the new method will require more computations and it could be possibly incompatible with privacy goals such as user differential privacy.

**Questions:**

My main questions are about the motivation behind leveraging SVD and the overall presentation of the work. as presented in the section above.

---

### Note · Authors · 2024-11-26

I have read and agree with the venue's withdrawal policy on behalf of myself and my co-authors.